# Incidence of Bloodstream Infection in Patients with Pulmonary Hypertension under Intravenous Epoprostenol or Iloprost—A Multicentre, Retrospective Study

**DOI:** 10.3390/ijms24076434

**Published:** 2023-03-29

**Authors:** Raquel Paulinetti Camara, Francisco das Neves Coelho, Natália Cruz-Martins, Patrícia Marques-Alves, Graça Castro, Rui Baptista, Filipa Ferreira

**Affiliations:** 1Cardiology Departament, Hospital Garcia de Orta, 2805-267 Almada, Portugal; 2Pulmonology Department, Hospital Nossa Senhora do Rosário, Centro Hospitalar Barreiro-Montijo, 2830-003 Barreiro, Portugal; 3Polyvalent Intensive Care Unit, Hospital Egas Moniz, Centro Hospitalar Lisboa Ocidental, 1349-019 Lisbon, Portugal; 4Faculty of Medicine, University of Porto, 4099-002 Porto, Portugal; 5Institute for Research and Innovation in Health (i3S), University of Porto, 4099-002 Porto, Portugal; 6Institute of Research and Advanced Training in Health Sciences and Technologies (CESPU), Rua Central de Gandra, 4585-116 Gandra, Portugal; 7TOXRUN—Toxicology Research Unit, University Institute of Health Sciences, CESPU, CRL, 4585-116 Gandra, Portugal; 8Pulmonary Vascular Disease Unit, Cardiology Department, Centro Hospitalar e Universitário de Coimbra, 3004-561 Coimbra, Portugal; 9Cardiology Department, Centro Hospitalar de Entre o Douro e Vouga, 4520-211 Santa Maria da Feira, Portugal; 10Faculty of Medicine, University of Coimbra, 3004-531 Coimbra, Portugal; 11ICBR—Institute for Clinical and Biomedical Research, Faculty of Medicine, University of Coimbra, 3004-531 Coimbra, Portugal; 12Clinical Academic Center of Coimbra (CACC), 3004-561 Coimbra, Portugal

**Keywords:** pulmonary arterial hypertension, bloodstream infection, survival, synthetic prostacyclin analogs

## Abstract

Intravenous synthetic prostacyclin analogs (iPCAs), such as epoprostenol, treprostinil and iloprost have been widely used for the treatment of pulmonary arterial hypertension (PAH). Despite having good outcomes, continuous infusion of iPCAs has been associated with some adverse effects. Bloodstream infection (BSI) is one of the most severe complications, although poorly recognized, especially under iloprost administration, which few studies have addressed. This study aimed to compare the BSI incidence rates between intravenous iloprost and epoprostenol administration. Patients with pulmonary hypertension (PH) functional class III or IV receiving intravenous iloprost or epoprostenol through Hickman catheter, between 2004 and 2019, were retrospectively selected from two PH treatment centers. From a total of 36 patients (13 for iloprost and 23 for epoprostenol), 75% (*n* = 27) fulfilled the PAH criteria, mainly belonging to the idiopathic group. Overall BSI rate was 1.5/1000 days of treatment (3.38 and 0.09/1000 days for iloprost and epoprostenol, respectively). Patients receiving iloprost were at a higher risk of developing BSI than those receiving epoprostenol (HR: 12.5; 95% CI: 1.569–99.092). A higher mortality rate from BSI was also identified in the iloprost group (*p* = 0.04). Twenty-seven patients developed BSI, with 92% of them requiring hospitalization. A total of 29 agents were found, 10 Gram-positive (mainly *Staphylococcus aureus*; *n* = 5) and 19 Gram-negative (mainly *Pseudomonas aeruginosa*; *n* = 6) bacteria. Iloprost administration was linked to a significantly higher incidence of BSI, worse prognosis, and more BSI-related deaths than epoprostenol. BSI due to Gram-negative, commensal, low-virulence bacteria was also higher in the iloprost group. In short, physicians should be aware when prescribing iPCA to guarantee their patients’ safety and best medical care.

## 1. Introduction

Pulmonary arterial hypertension (PAH) is a chronic condition characterized by proliferative vasculopathy in which structural changes of the pulmonary vascular lead to hemodynamic changes and right ventricular failure [1,2,3]. The currently approved therapies target three pathways involved in endothelial dysfunction—the prostacyclin and nitric oxide (NO) pathways, which are impaired in PAH patients, and the endothelin pathway, which is overactivated in these patients. In the last decades, several advances have been made in the therapeutic management of PAH; however, the underlying pathophysiological heterogeneity of PAH and its association with chronic and/or debilitating diseases impair treatment efficacy, resulting in higher morbidity and mortality rates [4].

Intravenous administration of synthetic prostacyclin analogs (iPCAs) is still the most effective therapy for severe PAH and has the most positive impact on patients’ survival [5]. The pharmacological effects of iPCAs are complex and include vasodilation of all vascular beds, inhibition of platelet aggregation, cytoprotection, antiproliferation and inotropism. These mechanisms improve pulmonary hemodynamics, ultimately resulting in symptom relief, increased functional capacity and improved survival [6]. iPCA therapy is approved in patients with PAH World Health Organization (WHO) functional class (FC) III and IV, however, parenteral administration of such drugs has relevant pharmacokinetic, pharmacodynamic and economic considerations that may limit their use [7]. The currently available iPCA formulations, such as epoprostenol, treprostinil and iloprost, are mainly limited by their extremely short half-life, requiring continuous infusion to be effective and prevent hemodynamic deterioration and even death. Among the iPCAs, iloprost has the longest half-life, which is relevant in case of accidental interruption of drug administration by catheter or pump malfunction. In addition, the use of such drugs is linked to serious adverse events, such as catheter obstruction, infection of the insertion site, bloodstream infection (BSI) and sepsis [8].

Previous studies have shown the association between prolonged administration of iPCA and the risk of BSI. Moreover, a sub-analysis of the REVEAL registry, which is the largest published series of patients with BSI associated with iPCAs to date, identified 1146 patients with PAH and BSI in 55 North American centers, with an overall incidence rate of 0.20 events per 1000 days of treatment [9]. Nevertheless, the incidence rate of BSI is still poorly characterized among studies, with variable results concerning the frequency of events and the prevalence of identified agents [10,11]. This scarcity of information is particularly striking regarding intravenous iloprost. Moreover, no previous studies have directly compared BSI rates between intravenous iloprost and epoprostenol therapy.

In this sense, we performed a retrospective longitudinal analysis of patients with PH to determine the incidence rate of BSI associated with intravenous iloprost and thermostable epoprostenol administration by a tunneled central venous catheter and a continuous infusion pump.

## 2. Results

### 2.1. Patients’ Characteristics

All patients received iPCA therapy with dose titration by other catheters until optimization was achieved and a one-lumen long-term tunneled catheter (Hickman^®^) was implanted. A total of 36 patients (13 for iloprost and 23 for epoprostenol) were enrolled (Table 1). The two groups, established according to the iPCA therapy administered, did not significantly differ in terms of demographic features (gender: *p* = 0.686, race: *p* = 0.525, age at diagnosis, *p* = 0.820). Most patients were Caucasian (94.4%), females (80.6%) and the median age at diagnosis was 43.0 (12–81) years. A total of 24 (66.7%) patients presented important comorbidities other than PH, and 1 patient from the iloprost group was pregnant during the treatment period. All patients received other PH drugs—75% were treated with endothelin receptor antagonists (ERA), 94% with phosphodiesterase type 5 inhibitors (PDE5i) and one patient was also under treatment with calcium channel blocker (CCB).

PH classification is depicted in Table 2. Altogether, 27 patients fulfilled the PAH criteria, mainly the idiopathic type (51.9%). Nine patients had other causes of PH: chronic thromboembolic pulmonary hypertension (CTEPH) (*n* = 7), myeloproliferative disorder (*n* = 1) and COPD (*n* = 1). In addition, five patients had more than one PH etiology but were grouped according to the main contributor to the disease: CTEPH and pulmonary restriction (*n* = 1), HIV plus systemic sclerosis and CTEPH (*n* = 1), CTEPH and chronic renal disease (*n* = 1), myeloproliferative disorder and COPD (*n* = 1) and COPD and sleep apnoea (*n* = 1).

### 2.2. Treatment Periods and BSI Incidence

There were no significant differences between groups regarding the time from PH diagnosis until iPCA treatment initiation (*p* = 0.987; Table 3). In total, patients had 22,273 days of iPCA therapy (median: 395.5 days), 10,599 days of iloprost treatment (median: 432 days) and 11,674 days of epoprostenol therapy (median: 329 days) (*p* = 0.267). Respecting iPCA administration via the Hickman catheter, patients had a total of 18,013 days of therapy (median: 306 days), 7685 days (median: 355) of iloprost and 10,328 (median: 288) of epoprostenol (*p* = 0.397).

The overall BSI rate was 1.5 per 1000 days of treatment (3.38 and 0.09/1000 days in the iloprost and epoprostenol groups, respectively). Patients who were under iloprost treatment were at a higher risk of developing BSI at any time than those who received epoprostenol (HR: 12.5; 95% CI: 1.569–99.092; Figure 1). Similarly, the incidence rate of BSI (27.8% vs. 2.8%; RR: 17.693 [95% CI: 2.543–123.1], *p* < 0.001) and the prevalence rate of multiple BSIs (19.4% vs. 0%, *p* < 0.001) were significantly higher in the iloprost group than in the epoprostenol group.

The mortality rate from BSI was significantly higher in the iloprost group, and no BSI-related deaths occurred in the epoprostenol group (23.1% vs. 0%, *p* = 0.04). All-cause mortality was not significantly different between both groups (30.8% vs. 39.1%, *p* = 0.974).

Although patients with BSI (Table 4) had more iPCA therapy and Hickman catheter days than those without BSI (median 675 vs. 291 iPCA days and 508 vs. 281 Hickman days, respectively), it was not statistically significant (*p* = 0.093 and *p* = 0.132, respectively).

### 2.3. Microbiologic Features in BSI Patients

Of the eleven patients with BSI, seven had multiple BSIs: four patients had one1 BSI, one patient had two, four patients had three, one patient had four and one patient had five BSIs. A total of 27 BSIs were diagnosed and 29 agents were identified from blood culture (two patients had two microorganisms identified simultaneously). Five patients (17.2%) had a CR-BSI diagnosis due to a positive catheter tip culture for the same agent identified in the blood culture.

Of the microorganisms identified, 10 (34.5%) were Gram-positive bacteria, all from the *Staphylococcus* family, with *Staphylococcus aureus* being the most frequent microorganism (*n* = 5), followed by *Staphylococcus epidermidis* (*n* = 3) and *Staphylococcus hominis* (*n* = 2). The remaining, 19 (65.5%) agents, were Gram-negative bacteria, with *Pseudomonas aeruginosa* being the most frequent (*n* = 6), followed *by Klebsiella oxytoca* (*n* = 3), *Klebsiella pneumoniae* (*n* = 2), *Leclercia adecarboxylata* (*n* = 2) and other low virulence bacteria (Table 5). Almost all BSIs (92.6%) required hospitalization for catheter removal and patient management.

## 3. Discussion

This study documented that continuous administration of intravenous iloprost was associated with a significantly higher incidence rate of BSI compared to epoprostenol. The number of studies addressing BSI in patients receiving intravenous iloprost is low and to the best of the authors’ knowledge, no previous studies have directly compared the incidence of BSI between iloprost and epoprostenol [10,12,13]. Nonetheless, data available so far have shown a BSI incidence rate with epoprostenol from 0.43 to 2.16 per 1000 days of iPCA therapy (studies >8 weeks in length) and with iloprost from 0.41 to 1.12 per 1000 days of iPCA therapy (Table 6) [5,9,10,12,13,14,15,16,17,18,19,20,21,22,23,24,25,26,27,28,29,30,31,32]. In the iloprost group, BSI also had a negative prognostic impact, with three BSI-related deaths in this group versus zero in the epoprostenol group. Such differences have been linked to the pH of the solutions used [25]. Unlike iloprost and treprostinil, which are stable with neutral pH, epoprostenol has to be reconstituted with an alkaline diluent (pH 12) to ensure its stability. Some in vitro microbiological studies have reported that both bacterial growth and biofilm production significantly varies within different pH solutions. For example, extremely alkaline solutions, such as those used to dilute epoprostenol, led to bacterial biofilm production inhibition of most *Staphylococcus aureus* strains [33]. Even moderate alkaline solutions (pH 8.5) showed a significant reduction in the adhesion capacity, biofilm production and even the viability of some *S. aureus* and *Staphylococcus epidermidis* strains [34]. Similarly, in vivo experiments where alkaline solutions were used resulted in a significant reduction in BSI incidence rates. In a study that compared diluted treprostinil with its usual neutral pH solvent and treprostinil diluted with epoprostenol alkaline solvent, the group with alkaline solvent presented a significantly lower BSI incidence than the group with neutral pH, which supports the hypothesis that an alkaline pH inhibits bacterial growth in the solution and systems of perfusion [35].

Among patients with BSI, Gram-negative bacteria were more common than Gram-positive bacteria (65% vs. 35%, respectively), with *Pseudomonas aeruginosa* (six isolations) and *S. aureus* (five isolations) being the most frequently isolated agent. Interestingly, we also reported a remarkable number of commensal agents not commonly associated with bacteremia in immunocompetent individuals. These bacteria included: *Acinetobacter iwoffii*, a Gram-negative bacillus that usually colonises the oropharynx, skin and perineum, whose BSI were rarely reported in the literature and only linked to disease in immunocompromised patients [36]; *Burkholderia cepacia* and *Ralstonia pickettii*, Gram-negative bacilli that produce a biofilm resistant to decontamination with injectable solutes, and appear as opportunistic agents in nosocomial infections [37,38,39,40,41]; *Delftia acidovorans*, a Gram-negative commensal bacillus from water and soil, that was detected as a BSI triggering agent in patients on haemodialysis or who had long-term catheter for chemotherapy [42]; *Klebsiella oxytoca*, a low-virulence Gram-negative bacillus, which colonises the human gastrointestinal tract, and that has been occasionally associated with bacteremia in immunocompetent adults [43]; and lastly, *Leclercia adecarboxylata*, a Gram-negative bacillus, which is taxonomically close to *Escherichia coli*, although of very low virulence, limited to systemic infections in diabetic or immunocompromised patients [44].

**Table 6 ijms-24-06434-t006:** Summary of BSI rate in patients under iPCAs (adapted from Sammut et al. [10] and Boucly et al. [14]).

Study	Duration	Total BSI (n)	BSI Global Rate *	Gram-Negative (%)
**Iloprost**				
Knudsen et al., 2011 [12]	2002–2009	9	0.41	N/A
Keusch et al., 2013 [13]	2000–2012	11	1.28	27
Sammut et al., 2013 [10]	2007–2012	15	0.65	60
Camara & Coelho et al. ^º^	2004–2019	26	3.38	67.9
**Epoprostenol**				
Rublin et al., 1990 [15]	8 weeks	N/A	0	N/A
Barst et al., 1996 [5]	12 weeks	N/A	1.16	N/A
McLaughlin et al., 1998 [16]	1994–1995	N/A	0.22	N/A
Badesch et al., 2000 [17]	12 weeks	N/A	0.43	N/A
Sitbon et al., 2002 [18]	1992–2001	N/A	0.55	N/A
McLaughlin et al., 2002 [19]	1991–2001	N/A	0.45	N/A
Oudiz et al., 2004 [20]	1987–2000	88	0.26	4
Barst et al., 2007 [21]	2003–2006	N/A	0.43	N/A
Akagi et al., 2007 [22]	1999–2005	21 6	0.89 0.1	0 0
Kallen et al., 2008 [23]	2004–2006	49	0.42	15
Hiremath et al., 2010 [24]	12 weeks	N/A	2.16	N/A
Kitterman et al., 2012 [9]	2006–2009	66	0.12	28.3
Rich et al., 2012 [25]	2009–2010	12	0.4	50
López-Medrano et al., 2012 [26]	1991–2012	7	0.12	0
Nagai et al., 2012 [27]	1998–2008	N/A	0.18	N/A
Chin et al., 2014 [28]	8 weeks	N/A	0.57	N/A
Sitbon et al., 2014 [29]	12 weeks	N/A	0.81	N/A
Frantz et al., 2015 [30]	2010–2012	N/A	0.2	N/A
Courtney et al., 2015 [31]	1999–2014	N/A	0.21	N/A
McCarthy et al., 2018 [32]	2000–2014	15	0.73	13.3
Camara & Coelho et al. ^º^	2004–2019	1	0.09	0

* per 1000 days of treatment. *Epo* epoprostenol. *Ilo* iloprost. º current study.

Previous studies have reported an abnormal proportion of these unusual agents in PAH patients under iPCAs [10,13,23,26]. Therefore, the occurrence of a significant number of BSIs caused by low-virulence agents in these patients leads to the hypothesis of a potential immunosuppressive effect associated with this long-term therapy. Epoprostenol is a synthetic formulation of prostacyclin or prostaglandin I2 (PGI2), a metabolite of araquidonic acid with the ability to connect to a specific IP receptor. Its stimulation leads to the elevation of intracellular cyclic AMP, which results in a remarkable anti-inflammatory effect. Considering this, it has been postulated that PGI2 may play a role in controlling innate and acquired immunity [45]. Other authors have suggested that endogenous prostanoids and synthetic derivatives exert an immunosuppressive effect [46,47]. Thus, the increasing body of indirect evidence allows us to speculate that the immunosuppressive effect of iPCAs may proceed from a class effect with heterogeneous profiles of immune activity suppression. However, the reduced number of publications in this field does not yet allow us to infer the immunological impact of long-term iPCA administration in vivo.

Nevertheless, besides considerations related to the drug itself and the associated pathogens, three relevant confounding factors may partly explain the higher rate of BSI in the iloprost group compared to the other case series listed here (Table 6). These factors were as follows: (1) promotion of patient participation and autonomy in catheter self-care procedures, (2) learning curve from the expert teams from both centers, and (3) development of new safety and aseptic techniques. Below, we briefly explain each of them:(1)There are only four specialized PH centers in Portugal, and the geographical distribution of patients impairs their ability to attend the hospital every 72 h to recharge their pumps. Additionally, primary care units are not prepared to manage these patients and devices properly. Thus, to ensure that the best medical care is provided to all patients without sacrificing their autonomy, the hospitals provide medication and the necessary material for self-preparation and administration of iPCAs, as well as regular training towards the asepsis maintenance in long-term catheters, management of home therapy, recognition of potential problems with the pump or other parts of the system and when to call hospital lines upon recognition of symptoms. The real impact of these measures on the prevalence of BSI is not fully elucidated, but it is reasonable to assume that intravenous therapy prepared by individuals who are not healthcare professionals may constitute an increased risk of contamination. Nevertheless, since this factor applies to patients from both therapy groups, this explanation itself seems to be insufficient to explain the discrepancy in BSI incidence rates found.(2)The learning curve for a period of >15 years in the two centers is another important factor that must be considered. Iloprost was the first iPCA available in Portugal in 2004, but only in 2014 did the thermostable epoprostenol become available for prescription in Portugal. Comprehensibly, patients received only iloprost for the first 10 years of iPCA national implementation and only when epoprostenol became available they could be offered that therapy. Moreover, it is relevant to consider that all patients in the epoprostenol group started their therapy and training with an expert team with 10 years of experience, thereby benefiting not only from the medical experience but also from the accumulated experience in patient education.(3)Since the introduction of iPCAs in Portugal, new measures have been implemented according to state-of-the-art techniques. Moreover, professionals have gained experience using strategies that can prevent the occurrence of infections, such as the use of a non-return-valve (closed hub) and waterproof connections [19,48]. Similarly, a teaching program and long-distant assistance via cell phone have been developed as an effort to better prepare and support patients to deal with unexpected events and to provide an immediate response to any doubts that might arise during their therapy management.

Looking at currently available published data (Table 6), the low incidence rate of BSI reported in this paper in the epoprostenol group is remarkable, and we believe that this finding might be due to the accumulated experience and improved practices of both centers.

This retrospective real-world study has some limitations. Only patients aged over 18 years were included, while other studies included patients >12 years of age. In addition, we included patients with multiple PH aetiologies (FC III or IV), requiring maximum medical therapy who received off-label iPCA, while most studies have specifically addressed patients with PAH.

Despite potential biases and limitations, the current study has also had a major strength. This is, to the best of our knowledge, the only study that directly compared BSI incidence rates between patients who received intravenous epoprostenol and iloprost.

## 4. Materials and Methods

### 4.1. Study Design

A retrospective cohort study was conducted, including patients with PH from two PH treatment centers treated with intravenous iloprost or epoprostenol between 1 January 2004 and 1 August 2019. This study was conducted in accordance with the Helsinki Declaration, and the study protocol was approved by the local ethics committee (approval number 7/2016).

### 4.2. Inclusion and Exclusion Criteria

Patients aged over 18 years old who were diagnosed with PH and precapillary PAH FC III or IV and received continuous iPCAs using the Hickman catheter for at least 24 h were included. According to the European Society of Cardiology/European Respiratory Society (ESC/ERS) guidelines, PAH is defined as: mean pulmonary artery pressure (mPAP) of >20 mmHg, pulmonary vascular resistance (PVR) of >2 Wood and pulmonary arterial wedge pressure (PAWP) of ≤15 mmHg [48].

Patients who did not receive iPCA using the Hickman catheter were excluded to avoid extra confounding factors.

### 4.3. Data Collection

Patients’ medical records were reviewed, and data related to age, gender, race, date of diagnosis, PH classification, management with other medical therapies for PH, comorbidities, occurrence of BSI, BSI-related microorganisms, catheter tip and peri-catheter exudate microbiology analysis (when applicable), date and cause of death were extracted.

Data related to the characteristics of therapy, including initiation and discontinuation dates of iPCA therapy, type of iPCAs used and date of vascular access implantation and removal were also collected.

### 4.4. Clinical Outcomes

The main outcome was the incidence rate of BSI between patients who received intravenous iloprost (Ilomedin, Bayer^®^, Leverkusen, Germany) and thermostable epoprostenol (Veletri, Actelion^®^ Pharmaceuticals US, Inc., South San Francisco, CA, USA) via a tunneled central venous catheter (Hickman^®^ catheter) and a continuous infusion pump (CADD Legacy^®^). Patients who received iloprost and epoprostenol were compared in terms of gender, race, comorbidities, PH classification, concomitant therapy for PH, age at diagnosis, time from diagnosis to the initiation of iPCA treatment, days of prostanoid treatment, days of treatment using the Hickman catheter, all-cause mortality rate and BSI mortality rate. Then, the BSI and non-BSI groups were compared with the same criteria as for the iPCA analyses. Microbiological characteristics of the isolated agents and the need for hospitalization were also evaluated.

The criteria for BSI diagnosis included at least one positive blood culture along with the presence of clinical signs indicative of systemic infection (fever [>38 °C], chills, hypotension and organ dysfunction) and absence of any identifiable septic focus other than the implanted vascular catheter [49].

The criteria for catheter-related BSI (CR-BSI) diagnosis included a BSI diagnosis, plus a positive culture of the catheter tip with the same microorganism identified from blood culture.

### 4.5. Statistical Analysis

Categorical variables are presented as absolute and relative frequencies, and continuous variables as mean and standard deviation, or median, interquartile ranges, and minimum and maximum values, when applicable. Data normality distribution was assessed using the Shapiro–Wilk test. Student’s *t*-test was used to compare the means of two normally distributed groups; when non-normally distributed, a nonparametric Mann–Whitney U test was utilized. The chi-square test or Fisher’s exact test was used to compare qualitative variables, as appropriate. A Cox proportional hazard model was established to determine time to BSI, and a Kaplan–Meier curve was generated. All data were analyzed using the Statistical Package for the Social Sciences (SPSS) software version 23, with an alpha set at 0.05.

## 5. Conclusions

In this study, a higher BSI incidence rate was reported in severe PAH patients under intravenous iloprost compared to those treated with intravenous epoprostenol. Iloprost therapy was also linked to a worse prognostic impact, as it was associated with more BSI-related deaths. Interestingly, besides *P. aeruginosa*, iloprost-related BSIs were mainly caused by commensal, low-virulence, Gram-negative bacteria which did not occur within epoprostenol therapy patients. The alkaline solvent of epoprostenol seems to play a protective role, helping to explain the differences found. Nonetheless, several confounding factors should be highlighted, as they can impact the results obtained: (1) the promotion of patient participation and autonomy in catheter self-care procedures, (2) the learning curve from the expert teams from both centers and (3) the retrospective study design with potential selection biases. In short, and although more studies are needed, the data obtained here emphasize the potential immunosuppressive role of intravenous prostacyclins, specially iloprost, and strongly support the worldwide use of epoprostenol, as it is recommended, to guarantee patients’ safety and the best medical care.

## Figures and Tables

**Figure 1 ijms-24-06434-f001:**
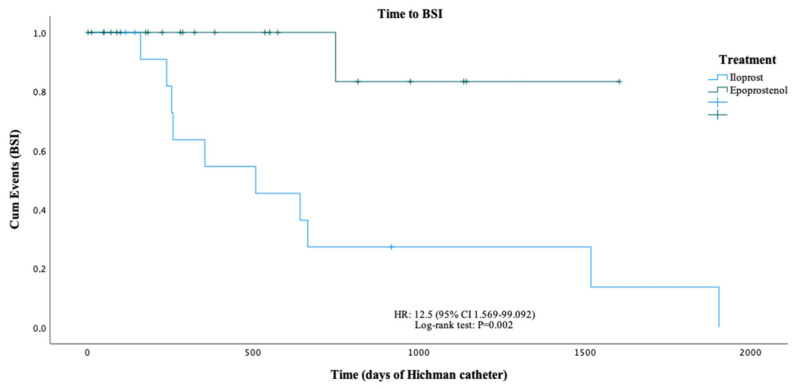
Analysis of BSI occurrence in PH patients treated with intravenous epoprostenol or intravenous iloprost.

**Table 1 ijms-24-06434-t001:** Patient characteristics.

Variables	Total (*n* = 36)		Iloprost (*n* = 13)		Epoprostenol (*n* = 23)		*p*-Value
Gender, *n* (%)							0.686
Male	7	(19.4)	3	(23.1)	4	(17.4)	
Female	29	(80.6)	10	(76.9)	19	(82.6)	
Race, *n* (%)							0.525
Caucasian	34	(94.4)	13	(100.0)	21	(91.3)	
African	2	(5.6)	0	(0.0)	2	(8.7)	
Age at diagnosis, median (min-max)	43.0	(12–81)	44.0	(16–77)	42.0	(12–81)	0.820
Comorbidities, *n* (%)							
Yes	24	(66.7)	9	(69.2)	15	(65.2)	0.806
PH therapy, *n* (%)							
ERA	27	(75)	13	(100.0)	14	(60.9)	**0.014**
PDE-5i	34	(94.4)	12	(92.3)	22	(95.7)	0.674
CCB	1	(2.8)	0	(0)	1	(4.3)	0.446

Bold: *p* < 0.05. PH pulmonary hypertension. ERA endothelin receptor antagonists. PDE-5i phosphodiesterase type-5 inhibitors. CCB calcium channel blocker.

**Table 2 ijms-24-06434-t002:** Classification of PH etiology in the study population (*n* = 36).

PH Aetiology, *n* (%)			
	1 PAH		27 (75.0)
		1.1 Idiopathic PAH	14 (38.9)
		1.2 Heritable PAH	5 (13.9)
		1.3 Drug- and toxin-induced PAH	1 (2.8)
		1.4 PAH associated with:	
		1.4.1 Connective tissue disease	3 (8.3)
		1.4.2 HIV infection	1 (2.8)
		1.4.3 Portal hypertension	1 (2.8)
		1.4.4 Congenital heart disease	2 (5.6)
		1.4.5 Schistosomiasis	0 (0)
		1.5 PAH long-term responders to CCB	0 (0)
		1.6 PAH with overt features of venous/capillaries (PVOD/PCH) involvement	0 (0)
		1.7 Persistent PH of newborn syndrome	0 (0)
	2 PH due to left heart disease		0 (0)
	3 PH due to lung disease and/or hypoxia		1 (2.8)
	4 PH due to pulmonary artery obstructions		7 (19.4)
	5 PH with unclear and/or multifactorial mechanisms		1 (2.8)
	More than one group classification *		5 (13.9)

* Patients already included in the main analysis of group etiology according to the major contributor to the disease. PH pulmonary hypertension. PAH pulmonary arterial hypertension. CCB calcium channel blockers. HIV: human immunodeficiency virus infection. PVOD pulmonary veno-occlusive disease. PCH pulmonary capillary haemangiomatosis.

**Table 3 ijms-24-06434-t003:** Characterization of treatment periods and the occurrence of BSI.

Variables	Total (*n* = 36)	Iloprost (*n* = 13)	Epoprostenol (*n* = 23)	*p* Value
Years of disease until starting iPCA, median (min-max)	4 (0–29)	4 (0–9)	4 (0–29)	0.987
Days of treatment, median (min-max)	395.5 (10–2651)	432 (117–2651)	329 (10–1638)	0.267
Days of Hickman, median (min-max)	306 (3–1904)	355 (117–1904)	288 (3–1604)	0.397
BSI (Hickman), *n* (%)	11 (30.6)	10 (76.9)	1 (4.3)	**<0.001**
Multiple infections, *n* (%)	7 (19.4)	7 (53.8)	0	**<0.001**
Deaths during iPCA therapy, *n* (%)				
All causes	13 (36.1)	4 (30.8)	9 (39.1)	0.974
BSI	3 (8.3)	3 (23.1)	0 (0)	**0.04**

Bold: *p* < 0.05. BSI bloodstream infection. iPCA intravenous prostacyclin analogs.

**Table 4 ijms-24-06434-t004:** Characterization of patients who developed BSIs (*n* = 36).

Variables	Total (*n* = 36)	No BSI (*n* = 25)	BSI (*n* = 11)	*p* Value
Gender, *n* (%)				0.559
Male	7 (19.4)	6 (24.0)	1 (9.1)	
Female	29 (80.6)	19 (76.0)	10 (90.9)	
Race, *n* (%)				0.861
Caucasian	34 (94.4)	23 (92.0)	11 (100)	
African	2 (5.6)	2 (8.0)	0	
Comorbidities, *n* (%)	24 (66.7)	16 (64.0)	8 (72.7)	0.715
Age at diagnosis (years), median (min-max)	43 (12–81)	45 (12–81)	36 (16–67)	0.342
Time from diagnosis to iPCA therapy (years), median (min-max)	4 (0–29)	4 (0–29)	4 (0–14)	0.520
Days of treatment, median (min-max)	395.5 (10–2651)	291 (10–2651)	675 (203–2059)	0.093
Days of Hickman, median (min-max)	306 (3–1904)	281 (3–1604)	508 (161–1904)	0.132

BSI: Bloodstream infection. iPCA intravenous prostacyclin analogs.

**Table 5 ijms-24-06434-t005:** BSI characterization.

Variables	*n* (%)
Total nº of BSIs during iPCA therapy, *n* = 27	
Multiple BSIs by iPCA cycle	7 (25.9)
2	1
3	4
4	1
5	1
Infections caused by more than one MO	2 (7.4)
Nº of microorganisms identified, *n* = 29	
Gram-positive	10 (34.5)
*Staphylococcus aureus*	5
*Staphylococcus epidermidis*	3
*Staphylococcus hominis*	2
Gram-negative	19 (65.5)
*Pseudomonas aeruginosa*	6
*Klebsiella oxytoca*	3
*Klebsiella pneumonia*	2
*Leclercia adecarboxylata*	2
*Acinetobacter iwoffii*	1
*Acinetobacter baumannii*	1
*Enterobacter cloacae*	1
*Burkholderia cepacian*	1
*Delftia acidovorans*	1
*Ralstonia pickettii*	1
Identification method	
Blood cultures	29 (100)
Hickman cateter tip (CR-BSI)	5 (17.2)
Catheter exudate	2 (6.9)
Hospitalization	25 (92.6)

BSI: Bloodstream infection. iPCA: intravenous prostacyclin analogs. MO: microorganisms.

## Data Availability

The data presented in this study are available on request from the corresponding author. The data are not publicly available due to privacy reasons.

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
