# Peer review of "Incidence of Bloodstream Infection in Patients with Pulmonary Hypertension under Intravenous Epoprostenol or Iloprost—A Multicentre, Retrospective Study"

_ijms, 2023, doi:10.3390/ijms24076434_

Round 1

Reviewer 1 Report

The methodology used to select the two groups under study and comparison are potentially biased and this severely limits the conclusions drawn

What is the biological plausibility of the results obtained?

Author Response

Dear Reviewer 1,

We kindly thank you for your suggestions to our work.

As suggested, we performed a full revision of our English grammar and syntax, which we hope you find acceptable. If you have any other questions or remarks regarding about our work, we would be glad to answer them.

Point 1: The methodology used to select the two groups under study and comparison are potentially biased and this severely limits the conclusions drawn

This was a longitudinal, real-world study that took place over a 15-year period. In the first 10 years of treatment (2004-2014), the only iPCA therapy available in Portugal for PH patients was iloprost. Since 2014, intravenous epoprostenol was authorized for medical prescription. Since epoprostenol was the recommended drug of choice to treat PH patients and isolated published works reported a lower incidence of infectious complications, all patients who began iPCA therapy were assigned to epoprostenol. Nevertheless, patients who were already under iloprost and had no infectious complications kept their therapy. For a period of 5 years, there was a superimposed use of both iPCA, depending on the time of diagnosis and patient evolution.

We are aware that this might be an important bias, that is why these aspects are clearly outlined in data interpretation. Nevertheless, we thoroughly reformulated our discussion section in order to clarify potential bias issues to the best of our ability.

Point 2: What is the biological plausibility of the results obtained?

In our discussion we try to address this relevant question:

  1. A potential immunosuppressive class effect: Previous studies in patients with PH and undergoing iPCA treatment have reported an abnormal number of BSI due to bacteria which are incapable of generating disease in immunocompetent hosts due to their low-virulence profile. In that sense, we raised the question whether there is an immunosuppressive class effect of iPCAs, as other authors previously did. The actual evidence doesn’t allow us to prove this theory and it will probably be difficult to do it in a near future since there are technical limitations to in vitro testing and ethical limitation to in vivo testing: iPCAs have an extremely short half-life which makes in vitro testing remarkably difficult; iPCAs clinical indication is mainly for severe PAH patients, a randomized clinical trial to prove if iPCAs are associated with more BSI, which would have to separate patients to receive best medical therapy (iPCA) vs only oral therapy wouldn’t be ethical in our point of view. This is why literature regarding this matter remains hypothetical and limited to considerations through cell pathway activation/inhibition.
  2. Different BSI rates between iloprost and epoprostenol: few studies pointed this might be due to differences in pH solutions preparation. Iloprost is stable with neutral pH solution, but epoprostenol must be reconstituted in alkaline pH solution. A number of studies have shown that Gram positive agents, namely S. aureus, cannot effectively produce biofilm under an alkaline environment. This might be an explanation to the difference found between the two drugs.

All available literature at time of paper submission regarding these findings is cited in our manuscript.

Reviewer 2 Report

1.      It will be interesting to know the background Hickman catheter BSI rates – although this can be difficult.

2.      Do we know how many are actually CRBSI?

3.      42-BSI was mainly due to gram-negative commensal, low-virulence bacteria. Not sure I will put it that way when the main results was pseudomonas.

4.      154- 1 patient had 4 and 1 patient had 5 BSI. All microorganisms  were  isolated  from  blood  culture  (17.2%  also  from  catheter  analysis  and  6.9%  from  1 catheter  exudates).  – What is the definition of catheter analysis and catheter exudates? this must be clearly defined

Author Response

Dear Reviewer 2,

We kindly thank you for questions and suggestions to our work.

As suggested, we performed a full revision of our English grammar and syntax, which we hope you find acceptable. If you have any other questions or remarks regarding about our work, we would be glad to answer them.

  1. It will be interesting to know the background Hickman catheter BSI rates – although this can be difficult.

Indeed, it is difficult to measure global Hickman catheter BSI rates due to a variety of reasons. As you may know, Hickman catheters are used for a very diverse number of reasons (chemotherapy, long-term antibiotic treatment, continuous parenteral nutrition, iPCA administration, etc.). Infection rate in such a diverse population depends both from the presence of the catheter, the drug administered and the underlying pathology of the patient.

We found an article describing Hickman line infection incidence in cancer patients, where the author considered the “normal range” of Hickman line infection to be somewhere between 0,3 to 8,0 infections per 1000 catheter days (Yip, C. & Rotstein, C., Hickman catheter-related infections in patients with cancer. International Journal of Antimicrobial Agents, 1998, 10(3), 181–189.), which appears to be a rather wide interval of infection, and a reference that may be more harmful than helpful in our discussion.

Nevertheless, we performed a literature review of BSI rates in PH patients under iPCA administered through Hickman catheter (Table 6).

  1. Do we know how many are actually CRBSI?

Yes. Five patients had a catheter tip analysis positive to the same agent identified in blood cultures.

We realized the terms BSI and CR-BSI were used interchangeably throughout the manuscript, which is, in fact, inaccurate. Therefore, we reviewed that it in this new version.

  1. 42-BSI was mainly due to gram-negative commensal, low-virulence bacteria. Not sure I will put it that way when the main results was pseudomonas.

You are correct. We replaced the sentences containing this imprecision.

  1. 154- 1 patient had 4 and 1 patient had 5 BSI. All microorganisms were isolated from blood culture (17.2% also from catheter analysis and 6.9% from 1 catheter exudates). – What is the definition of catheter analysis and catheter exudates? this must be clearly defined

You are correct.

To clarify this, in the Method section a section named “Definitions” was added where BSI and CR-BSI were clearly defined.

Inconsistencies in denomination have been cleared out from the whole manuscript to the best of our ability.

In the discussion section, when we discuss the limitations of our work, we clarify that there were logistical limitations that had a negative impact in determining the diagnosis of a CR-BSI (mostly due to our microbiology labs not being able to guarantee 2 out of 3 criteria for CRBSI when we consider IDSA guidelines regarding this matter).

Round 2

Reviewer 1 Report

The authors have made a reasonable attempt to justify methodology but the basic flaws remain and compromise the validity of the conclusions.

I dont feel this manuscript adds significantly to other published work

Author Response

Dear reviewer,

Thank you for taking your time to read again our manuscript and replay.

We understand your concerning about methodology. Indeed, this is not a randomized controlled trial, but a real-world retrospective observational study, and we cannot change its nature. As you certainly might know, the most important regulatory organizations, as FDA and EMA, are accepting real-world investigation (even retrospective) to approve new drugs and new drug indications, as you may see below from the FDA site transcription. We are not aiming to approve a new drug or indication, but in this world of constant evolution, older drugs are being rescued for new indications, or sometimes even for the same indication, so it is important to gather the much information that we can for every drug. Take for instance the use of tobacco cessation drugs – varenicline was one of the most important drugs for many years, but suddenly it was out of market and citizine took its place as an excellent new drug, when, in fact, it already existed for many years and no one ever paid attention to it (even if it was already addressed in the European Guidelines for Smoking Cessation since 2012). What we want to say is – older drugs might become new drugs and iloprost has some benefits, like its longer half-life comparing with epoprostenol and treprostinil. There are few studies reporting iv iloprost BSI rates, so we consider our contribution important to enrich literature. Plus, there are no studies comparing the BSI rates from the two drugs and here we compared ilo and epo administered in the same hospital by the same team, even though patients were not randomized. This is not an ideal methodology, we agree on that, but our results are a blind spot in this area, are independent from any pharmaceutical and do have statistical significance. We do consider this a paper worth to share with our pairs.

Nevertheless, we are aware of our limitations, that’s why we dedicated so many lines of our manuscript to discriminate limitations and confounding factors, as those you express. We can’t change the methodology because its retrospective, as you are aware, and since iloprost is now rarely used in PH, it is not expected that a strong study emerges soon. This is perhaps one of the last papers of this iloprost era in PH (although it might rise in the near future). The point we bring back to you is if you think scientific community benefits more from no data at all about this issue than from a well-documented limited data.

Still, our paper it’s not limited to our findings. We did an extensive revision of the literature regarding both drugs, which is easily accessible for citation in our table 6, and also, we performed a review of the uncommon microorganisms found, which is also interesting for readers and we didn’t find it altogether in other articles with such detail.

We hope we convinced you our paper strengths prevail over limitations and that this is worth to publish. Do you still have any suggestion about what we can actually improve?

Thank you again for your time and dedication.

“Real-world data (RWD) and real-world evidence (RWE) are playing an increasing role in health care decisions. (…) The 21st Century Cures Act, passed in 2016, places additional focus on the use of these types of data to support regulatory decision making, including approval of new indications for approved drugs. (…) Real-world evidence is the clinical evidence regarding the usage and potential benefits or risks of a medical product derived from analysis of RWD. RWE can be generated by different study designs or analyses, including but not limited to, randomized trials, including large simple trials, pragmatic trials, and observational studies (prospective and/or retrospective).” https://www.fda.gov/science-research/science-and-research-special-topics/real-world-evidence

Round 3

Reviewer 1 Report

Ive already expressed my views